# Bridging Inequity Gaps in Healthcare Systems While Educating Future Healthcare Professionals—The Social Health Bridge-Building Programme

**DOI:** 10.3390/ijerph20196837

**Published:** 2023-09-27

**Authors:** Gitte Valentin, Claus Vinther Nielsen, Anne-Sofie Meldgaard Nielsen, Merete Tonnesen, Kristina Louise Bliksted, Katrine Tranberg Jensen, Karen Ingerslev, Thomas Maribo, Lisa Gregersen Oestergaard

**Affiliations:** 1DEFACTUM, Central Denmark Region, 8000 Aarhus, Denmark; claus.vinther@rm.dk (C.V.N.); anmne9@rm.dk (A.-S.M.N.); merete.tonnesen@rm.dk (M.T.); thomas.maribo@rm.dk (T.M.); lisaoest@rm.dk (L.G.O.); 2Department of Public Health, Aarhus University, 8000 Aarhus, Denmark; 3Department of Clinical Social Medicine and Rehabilitation, Gødstrup Regional Hospital, 7400 Herning, Denmark; 4Social Sundhed (Social Health), 8000 Aarhus, Denmark; klb@socialsundhed.org (K.L.B.); ktj@socialsundhed.org (K.T.J.); ki@socialsundhed.org (K.I.); 5Department of Public Health, Copenhagen University, 1353 Copenhagen, Denmark

**Keywords:** inequity, inequality, deprivation, healthcare, student volunteer, complex intervention, programme theory

## Abstract

Social inequity in healthcare persists even in countries with universal healthcare. The Social Health Bridge-Building Programme aims to reduce healthcare inequities. This paper provides a detailed description of the programme. The Template for Intervention Description and Replication (TIDieR) was used to structure the description. The programme theory was outlined using elements from the British Medical Research Council’s framework, including identifying barriers to healthcare, synthesising evidence, describing the theoretical framework, creating a logic model, and engaging stakeholders. In the Social Health Bridge-Building Programme, student volunteers accompany individuals to healthcare appointments and provide social support before, during, and after the visit. The programme is rooted in a recovery-oriented approach, emphasising personal resources and hope. The programme finds support in constructs within the health literacy framework. Student volunteers serve as health literacy mediators, supporting individuals in navigating the healthcare system while gaining knowledge and skills. This equips students for their forthcoming roles as healthcare professionals, and potentially empowers them to develop and implement egalitarian initiatives within the healthcare system, including initiatives that promote organisational health literacy responsiveness. The Social Health Bridge-Building Programme is a promising initiative that aims to improve equity in healthcare by addressing individual, social, and systemic barriers to healthcare. The programme’s description will guide forthcoming evaluations of its impact.

## 1. Introduction

The social gradient in health is a growing concern worldwide, with persistent and increasing social inequalities in some countries [1,2]. In Europe, these inequalities lead to an average reduction in life expectancy of 5–10 years, and a decrease in disability-free life expectancy by 10–20 years [3]. The World Health Organization defines social determinants of health as “the conditions in which people are born, grow, work, live and age, and the wider set of forces and systems shaping the conditions of daily lives” [4]. 

These determinants stem from social, economic, and political mechanisms, which subsequently give rise to a set of socioeconomic positions, whereby populations are stratified according to education, occupation, income, gender, ethnicity and other factors. In turn, these socioeconomic positions shape specific determinants of health status depending on individual’s place within social hierarchies. According to their respective social position, individuals experience differences in exposure and differences in vulnerability to health problems [5]. The underlying social determinants of health inequities operate through a set of intermediate determinants of health to shape health outcomes. These intermediate determinants include housing, financial means, environmental stressors, social support, and health behaviour (i.e., poor nutrition, physical inactivity, smoking, and alcohol consumption) [5]. In essence, the roots of social inequalities in health are multifaceted, and predominantly exist outside the domain of the healthcare system. Addressing health inequalities thus requires action on inequalities in wider social determinants outside the healthcare system [6].

The healthcare system does, however, play an important role in mitigating the consequences of social inequalities. The healthcare system can both exacerbate existing inequalities, and actively contribute towards reducing social inequality through the way in which activities are organised [7]. Ensuring universal access to high quality care and a focus on equitable outcomes, is thus central to challenging health inequities [7].

Even in publicly funded healthcare systems like those in Canada, the UK, and Scandinavia, healthcare inequities persist [8,9]. Disadvantaged populations with greater healthcare needs often receive less care than advantaged populations with lower needs, resulting in unjust health inequalities [10]. Disadvantaged populations are less likely to have a general practitioner, receive preventive and secondary care, and often resort to using the emergency department and report negative experiences of care [11,12,13,14,15]. Individuals with lower socioeconomic position often have greater needs for support accessing care, but do not experience adequate social support from network, and healthcare professionals often find it challenging to meet their needs [9,16,17,18].

To address these issues, the Social Health Bridge-Building Programme was established with the objective of reducing healthcare inequity while educating future healthcare professionals [19]. The programme involves healthcare student volunteers, such as future doctors, nurses, or psychologists, accompanying individuals to healthcare appointments and providing social support before, during, and after the visit. These volunteers, referred to as “bridge-builders,” aim to bridge the gaps between civil society and the healthcare system, enhancing accessibility for individuals who struggle to navigate it. [19]. The target population for the Social Health Bridge-Building Programme is individuals in vulnerable situations, who require support from a bridge-builder to remember, maintain, or derive better outcome from their healthcare visits. The term “individuals in vulnerable situations” is employed to emphasise that vulnerability is a dynamic phenomenon with multiple dimensions that mutually influence each other. Vulnerability can be transient or persistent, and its degree can vary depending on the living conditions individuals experience [20]. In this context, vulnerability is influenced by both individual resources, including socioeconomic position and social networks, and the challenges encountered throughout life, such as mental and physical illness [21]. Hence, the programme is not limited to disadvantaged individuals but includes a broader population, and no referral is necessary [19]. The Social Health Bridge-Building Programme was initiated in 2013 by a Danish NGO called Social Health, starting on a small scale in one city. By 2023, it has expanded nationwide, with headquarters in the five largest university cities. Currently, the programme has 12 employed coordinators and 200 healthcare student volunteers who accompany individuals to 2500 healthcare appointments each year. While the programme has continuously evolved based on input from stakeholders, including individuals in vulnerable situations, healthcare professionals, policymakers, and students, the proposed pathways from programme activities to increase equity in healthcare have not previously been elucidated. Detailed published descriptions provide a structured approach to analyse, understand, and evaluate programmes and interventions by identifying and examining the assumptions and causal pathways that underlie their effectiveness. Furthermore, they enable implementation in other settings, allow other researchers to replicate research findings, and enable reviewers to synthesise existing evidence [22]. 

Hence, the aim of this paper is to provide a detailed description of the key elements of the Social Health Bridge-Building Programme, and to unfold the programme theory.

## 2. Materials and Methods

The Template for Intervention Description and Replication (TIDieR) was used as a structural framework for the description of the key elements of the Social Health Bridge-building Programme [22]. The programme theory, i.e., the rationale and theory essential to the programme (item 2 and item 3 in the TIDieR checklist), was unfolded based on elements from the British Medical Research Counsel’s (MRC) framework and included (1) identification of barriers to healthcare for individuals in vulnerable situations (problem analysis), (2) identification and synthesis of the evidence base, (3) description of the theoretical foundation, (4) development of a logic model, and (5) engagement of stakeholders [23]. 

### 2.1. Element 1: Identifying Barriers to Healthcare 

A problem analysis was conducted in order to gain a profound understanding of the public health issue targeted by the Social Health Bridge-Building Programme. Hence, barriers to healthcare among individuals in vulnerable situations were identified and examined. The identification of barriers drew upon a recent systematic review conducted by the Danish Health Authorities [24], and ethnographic fieldworks carried out among different stakeholders of the Social Health Bridge-Building Programme using interviews and participant observation [25]. 

### 2.2. Element 2: Identification and Synthesising the Evidence Base

A comprehensive evidence synthesis was conducted to explore the existing evidence regarding the engagement of healthcare student volunteers in improving social equity in healthcare. The literature searches and study selection procedures followed the guidelines outlined by the Preferred Reporting Items for Systematic Review and Meta-Analysis (PRISMA) statement [26]. Searches were conducted in the PubMed, CINAHL, Scopus, and ProQuest databases, covering the period from inception to the final week of February 2023. The literature search strategy was developed collaboratively with a research librarian. The search strategy developed for PubMed is presented in Appendix B. A broad set of search terms was used for each concept combined with the Boolean operator OR to increase sensitivity within the concepts. All three databases were searched using a combination of text words and standardised subject terms (e.g., Medical Subject Headings (MeSH)). The search was not restricted by language or publication format. To ensure comprehensive coverage, the reference lists of eligible studies were examined. All records obtained from the literature search were imported into the Covidence platform. Initially, all duplicates were removed by Covidence. The remaining records were screened independently by two authors at title and abstract level against predefined eligibility criteria. Inclusion criteria for the study encompassed published peer-reviewed qualitative or quantitative studies that examined the engagement of healthcare student volunteers as a means to improve social equity in access to and benefit from the healthcare system for individuals in vulnerable situations. Records were excluded if their title and abstract did not fulfil the eligibility criteria. Full text papers were obtained for all remaining records. Inclusion was agreed upon via a consensus and, if necessary, through discussion with a third co-author.

### 2.3. Element 3: Description of the Theoretical Foundation

Theory offers a structural foundation for comprehending the fundamental causal connections and mechanisms that contribute to a specific problem or challenge. Through the identification of pertinent theoretical models or concepts that are relevant to the context of the intervention, one can gain deeper insight into the efficacy of specific approaches and the factors involved [27]. Consequently, the core theoretical concepts of the Social Health Bridge-Building Programme were elucidated to reinforce the scientific rationale, and foster a deeper understanding of the mechanisms through which the intended outcomes can be achieved.

### 2.4. Element 4: Development of a Logic Model

A logic model was formulated to systematically present and visualise the proposed pathways, starting from the resources and activities within the programme to the attainment of increased equity in healthcare [28]. This model emphasises the necessary resources (physical, economic, and human) as well as the core activities undertaken within the programme.

### 2.5. Element 5: Engagement of Stakeholders

The programme description was elaborated by the research team (GV, CVN, ASMN, MT, TM and LGO) in close collaboration with board members and the CEO of the Social Health Bridge-Building Programme (KLB, KTJ and KI). Initially, a draft version of the TIDieR checklist, including the programme theory, was conducted at a workshop with the representatives of the Social Health Bridge-Building Programme. Then, the description underwent multiple refined iterations through a collaborative effort between the research team and the stakeholders. The elaborations were based on inputs from the literature review and the qualitative research. Hence, empirical evidence and theory were employed to explicate and substantiate the assumptions regarding the causal pathways through which the desired effects of the programme are achieved. Lastly, the refined programme theory was visualised in a simplified logic model.

## 3. Results

The programme theory (element 1–4) is explained in the following sections. The programme theory predominantly corresponds to item 2 (why) and item 3 (what) of the TIDieR checklist. The remaining items in the TIDieR checklist are available in Appendix A. 

### 3.1. Element 1: Identifying Barriers to Healthcare

The systematic review applied to identify barriers to healthcare for individuals in vulnerable situations encompassed 14 quantitative and 10 qualitative peer-reviewed studies, published between 2010 and 2020 [24]. These studies originated from various countries, including the UK, the U.S., France, Sweden, Germany, Canada, and Australia. To insure generalisability of the results into a Danish healthcare setting, findings from the review was supplemented with findings from ethnographic fieldworks among different stakeholders of the Social Health Bridge-Building Programme [25]. The review revealed barriers at multiple levels, indicating that interventions targeting these barriers should address multiple levels as well. The identified barriers were categorised into (1) intrapersonal (individual), (2) interpersonal (social), and (3) healthcare system; they are presented in a socio-ecological model, inspired by the work of Dahlgren and Whitehead [29] (Figure 1).

At the intrapersonal level, barriers to healthcare include unhealthy living conditions such as poor housing and environmental stress [16], low health literacy [30,31,32], limited economic resources such as lack of funds for transportation, medication, and user-paid healthcare services like physiotherapy or dental care [30,32,33,34], insufficient social support [16,33,34], and cultural differences such as language barriers. At the interpersonal level, barriers include communication difficulties stemming from healthcare professionals’ inadequate interpersonal skills and use of medical jargon, as well as individuals’ lack of trust due to previous negative experiences and feelings of embarrassment, and social distance resulting from healthcare professionals’ prejudices, labelling, lack of empathy, and disrespect, as well as differing choices and perspectives [16,34,35,36,37,38,39]. At the system level, barriers include (1) lack of resources in the healthcare system and the hereof shortage of healthcare professionals resulting in limited time to address the complex needs of individuals in vulnerable situations [16,33,39,40]; (2) healthcare system complexity, making it challenging for people in vulnerable situations to navigate and comprehend the necessary processes and procedures [16,25,30,33,41]; and (3) difficult access to primary care physician and specialised treatment [25,30,31,41,42,43,44].

### 3.2. Element 2: Identification and Synthesising the Evidence Base

The systematic review on interventions to reduce healthcare inequity through student volunteers yielded 482 references, after removing the duplicates. These references were screened for eligibility and 445 records were excluded. A total of 37 papers were read in full-text of which 30 were excluded. The reasons for exclusion were (1) study design (no intervention/programme presented in the paper, or expert opinion) (n:10), (2) intervention (the intervention was not relevant, e.g., interventions aiming to reduce social isolation among older people during the COVID-19 pandemic or interventions aiming to prevent preschool obesity) (n:9), (3) population (not individuals in vulnerable situations) (n:3) or (4) providers (not healthcare student volunteers) (n:4). In addition, studies that were included in the systematic review by Wilson et al. [45] were excluded to avoid cohort overlap (n:2). A total of seven papers were included: one systematic review [45] and six primary studies [25,46,47,48,49,50]. The flow chart of study inclusion and exclusion is available in Appendix C. The characteristics of the included studies are presented in Table 1. The vast majority of studies originated in a North American setting, and most were related to clinics involving solely medical students. One paper from our own research team, a qualitative study, covered the Social Health Bridge-Building Programme [25], while the remaining covered clinical exposure in student-run or shelter-based health clinics [45,46], or screening of social needs in community resource programs [47,48,49,50]. 

The impact of clinical exposure in student-run or shelter-based health clinics was explored in two papers [45,46]. Wilson et al. (2023) reviewed 92 studies, and found that participation in student-run clinics improved students’ learning outcomes, including clinical skills (e.g., interpersonal communication skills and experience in managing language barriers) and empathy for underserved patients [45]. Asgary et al. (2016) concluded that clinical exposure in shelter-based clinics, combined with active faculty precepting, enhanced medical students’ knowledge, attitudes, and skills to address healthcare needs among homeless individuals [46]. 

Screening for health related social needs was covered in four studies [47,48,49,50]. In the Community Navigator Programme, healthcare student volunteers make follow-up calls to patients referred to community resources to facilitate and evaluate connection to these resources (e.g., access to affordable medications or specialty care, food insecurity or financial assistance) [47,50]. In the Medical Student Advocate programme, medical students are trained to assess individuals’ social needs (e.g., child care, food, or transportation) and work collaboratively with the Patient-Centred Medical Home team to address the identified concerns [49]. Both programmes were found to enhance students’ empathy towards patients as well as their understanding of social determinants of health, and interdisciplinary collaboration [47,49]. Furthermore, students developed broader interpersonal and communication skills needed to become future health professionals [46]. In the Highland Health Advocates programme, undergraduate volunteers located at the Emergency Department (ED) help patients to navigate public resources and provide onsite legal and social work referrals [48]. The authors found that among the individuals who had received help from a student volunteer more were linked to a resource (59% vs. 37%) and a medical home (92% vs. 76%.) one month after visiting the emergency department. At 6 months, 75% felt that the Highland Health Advocates programme was helpful and more HHA subjects had a doctor (93% vs. 69%). No difference was found in ED utilisation, primary need resolution or self-reported health status [48].

### 3.3. Element 3: Description of the Theoretical Foundation

The proposed pathways, from accompanying individuals to healthcare appointments to increased social equity in healthcare, finds support in theoretical constructs within the health literacy framework. Health literacy is defined as “the combination of personal competencies and situational resources needed for people to access, understand, appraise and use information and services to make decisions about health” [51]. Evidence shows that health literacy mediates the relationship between socioeconomic position and health at the individual level and social inequity in health at the system level [52]. Hence, efforts that increases health literacy at the individual level or reduce the consequences of low health literacy can contribute to reduced inequity in health [51]. 

In recent years, the literature has drawn attention to the intersection between health literacy and social context acknowledging that an individual’s health literacy skills are supplemented by those of others (including social network and health professionals) [53]. The construct Distributed Health Literacy concerns the way in which individuals draw upon social networks to share knowledge and understanding, assess and evaluate information, communicate with health professionals, and support decision making [54]. Those who passes their health literacy skills to others act as health literacy mediators, as they support others in becoming more health literate to manage their conditions [53]. Bridge-builders may ameliorate the adverse impacts of low health literacy among individuals in vulnerable situations who have a scarce social network by providing Distributed Health Literacy. In line with the Distributed Health Literacy Pathway Model [54], bridge-builders acts as health literacy mediators by (1) providing support and motivating the individual to become more active in the appointment (e.g., by preparing questions for discussions in the healthcare appointment), (2) providing support with way-finding to the location of the appointment as well as support with communication during appointment (listening to consultations, acting as note-takers or provide some input to the healthcare professional during the appointment), and (3) providing support after the appointment with processing, understanding and evaluating the information that was provided during the appointment. 

Organisational Health Literacy Responsiveness is another construct of relevance to the Social Health Bridge-Building Programme within the health literacy framework. Organizational Health Literacy Responsiveness pertains to “the way in which services, organizations and systems make health information and resources available and accessible to people according to health literacy strengths and limitations” [53]. It encompasses the organisation’s ability to deliver clear and comprehensible health information, facilitate communication and shared decision-making with patients or service users, and foster an environment that supports health literacy [53]. The Social Health Bridge-building Programme is expected to facilitate Organisational Health Literacy Responsiveness by increasing future healthcare professional’s knowledge of health literacy, and raising awareness of the barriers within the healthcare system. 

Mandatory teaching of bridge-builders based on a recovery-oriented approach is a fundamental element of the Social Health Bridge-building Programme [55,56]. The bridge-builders are trained in the mind-set of actively promoting and carrying the hope for individuals, thus reinforcing their belief in the capacity to navigate challenges and successfully overcome barriers. Furthermore, this training aims at enabling the bridge-builders’ capacity to perceive individuals from a holistic standpoint, considering them as unique individuals rather than merely symptoms or diagnoses.

### 3.4. Element 4: Development of a Logic Model

The logic model (Figure 2) visualises the core resources, activities and output as well as the intended outcomes of the Social Health Bridge-Building Programme.

#### 3.4.1. Resources, Activities, and Outputs

The key ingredients in the Social Health Bridge-Building Programme are operation of the Advisory Hotline, accompaniment to healthcare appointments, recruitment, training and supervision of bridge-builders and lobbying for social equity in healthcare. 

The main aim of the programme’s Advisory Hotline is to connect an individual in need of accompaniment with a bridge-builder. The Advisory Hotline is staffed by employed coordinators, all of whom have backgrounds in social work, communication, relationship-building and/or healthcare education. The contact is established when the individual in need of accompaniment or more often someone close to the individual (e.g., social workers, caregivers or relatives) reaches out to the Advisory Hotline. In addition, the Social Health Bridge-Building Programme also conducts outreach work in different community settings. 

The bridge-builder’s primary responsibility is to provide support and assistance to individuals before, during and after healthcare appointments. Before the appointment, the bridge-builder meet the individual at a designated location (e.g., the individual’s residence, a nursing facility, or a shelter). Following an initial dialogue regarding the individual’s expectations and requirements related to the healthcare appointment, the bridge-builder accompanies the individual to the appointment, utilizing transportation methods such as taxis, buses, or other means of transport. During the appointment, the bridge-builder participates in the appointment or waits as requested by the individual. Afterward, the bridge-builder accompanies the individual to their place of residence, discussing the appointment if necessary. If authorised by the individual, pertinent information concerning the outcome of the healthcare appointment may be shared with their caregivers or relatives. 

To become a bridge-builder, healthcare students are required to complete a 20 h foundational training course on social determinants on health, communication, and boundary setting. The course teachers are individuals with educational background in social work and recovery, some with personal experiences of social vulnerability. The modules contain elements of theoretical, practical (case-based) and experience-based (role-play) knowledge. The bridge-builders are briefed by the employed coordinator before an assignment and debriefed afterwards. Additionally, ongoing supervision is provided by external supervisors to all bridge-builders to increase their ability to interact with individuals in vulnerable situations. During supervision, the bridge builders reflect on the experiential knowledge and expertise they bring from their experiences in healthcare accompaniment, and act as co-reflectors on each other’s experiences. 

In order to increase awareness and utility as well as funding for the Social Health Bridge-Building Programme, the employed coordinators extensively lobby with the community, healthcare system, and health educational institutions. 

#### 3.4.2. Expected Outcomes

In the logic model, the expected outcomes of the programme are delineated as intermediate, short-term (1–3 years), and long-term outcomes (4–6 years) across three distinct levels: (1) individuals in vulnerable situations, (2) bridge-builders, and (3) the healthcare system. The intermediate outcomes correspond to the underlying mechanisms of the intervention. Lastly, the model outlines the projected societal impact of the programme. 

The Social Health Bridge-Building Programme aims to enhance equity in healthcare by addressing various barriers at multiple levels. At the individual level, receiving accompaniment and support for healthcare appointments is expected to contribute to increased trust in the healthcare system, improved health literacy, or increased compensatory Distributed Health Literacy, and improved social support. These intermediary outcomes are expected to yield several short-term effects at the individual level, such as a more responsive utilisation of general practitioners, improved adherence to healthcare appointments, and reduced un-planned acute hospital visits. Furthermore, it is anticipated that the programme can enhance communication with healthcare professionals and self-efficacy. Consequently, these short-term outcomes are expected to result in improved health outcomes for the individuals.

At the bridge-builder level, accompanying individuals provides bridge-builders with valuable insights into recognising and addressing social determinants of health, and enables them to perceive the individuals from a holistic standpoint beyond their diagnosis. Furthermore, bridge-builders gain experience in effectively communicating and establishing respectful relationships with individuals in vulnerable situations, thereby enhancing their professional competencies to improve and cater to the individuals’ health literacy. As a short-term outcome, the acquired competencies are expected to enhance the bridge-builder’s understanding of the lived reality of individuals in vulnerable situations. These competencies are also anticipated to have a positive impact on appointments and other social circumstances, as they improve the bridge-builder’s communication skills, empathy, and respect when interacting with individuals. Furthermore, the improved competencies and experience with the individuals’ lived lives may help to reduce stigmatisation and prejudicial experiences within appointments or similar settings for this target group. As a result of these short-term outcomes, bridge-builders potentially become better equipped to meet the complex needs of individuals living in vulnerable situations in their future healthcare careers. 

At the healthcare system level, the Social Health Bridge-Building Programme is anticipated to promote organisational health literacy responsiveness by enhancing the understanding of health literacy among future healthcare professionals. Additionally, it aims to raise awareness of barriers within the healthcare system through extensive advocacy endeavours directed at various stakeholders, such as policymakers, healthcare system leaders, citizens, and both current and future health professionals.

Overall, the changes at the three levels have the potential to result in increased social equity in access to and benefits from the healthcare system.

## 4. Discussion

### 4.1. Summary of Overall Findings

The Social Health Bridge-Building Programme aims to address healthcare inequities in universal healthcare systems while educating future healthcare professionals. This paper provides a comprehensive description of the programme and its theoretical foundation, laying the groundwork for evaluations and potential implementation in different settings. The programme targets the barriers to healthcare by providing social support and assistance in navigating the healthcare system. It offers students first-hand exposure to the social determinants of health. The theoretical framework is based on a recovery-oriented approach, emphasising personal resources and hope. The student volunteers serve as health literacy mediators, supporting individuals in navigating the healthcare system while gaining knowledge and skills. This equips students for their forthcoming roles as healthcare professionals, and potentially empowers them to develop and implement egalitarian initiatives within the healthcare system, including initiatives that promote organisational health literacy responsiveness. Ethnographic research supports the role of student volunteers as substitutes for absent family or social networks, and it highlights the insights gained from accompanying individuals to healthcare appointments. 

### 4.2. Implications for Policy and Practice 

Healthcare systems worldwide are facing significant challenges as a result of the profound demographic shift characterised by a growing population of individuals over the age of 80, and a decline in the number of healthcare professionals and caregivers [57]. Amidst this situation, the involvement of family caregivers is being increasingly acknowledged as a valuable resource for both patients and the healthcare system [58]. However, this reliance on family caregivers presents difficulties for individuals lacking a supportive social network. In the coming years, patient advocacy and navigation services may play an important role in offering guidance and support to these individuals. Notably, non-governmental organisations like the Social Health Bridge-Building Programme have shown promise in bridging the gap for patients lacking resourceful caregivers, thus ensuring equitable care and support. However, patient advocacy and navigation services do not solve the inequity challenges faced by the healthcare system. Achieving a more egalitarian healthcare system requires active and continuous endorsement from policymakers and healthcare leaders. These leaders can leverage the skills and competencies of former bridge-builders by appointing them as ambassadors in their future careers, promoting equity and advocating for change.

Currently, the Social Health Bridge-Building Programme is being implemented in multiple cities in Denmark, with future aspirations for further national and international expansion. The successful and sustainable implementation of the programme relies on a well-structured organisation, including programme coordinators in the specific setting as well as a strong administration led by a CEO, who is backed by solid board. Additionally, support and endorsement from a diverse range of stakeholders are crucial for ensuring the programme’s sustainability. Firstly, sufficient funding is necessary to cover the expenses related to the Advisory Hotline, as well as the recruitment, education, and supervision of student volunteers. Secondly, close collaboration with universities and other healthcare educational institutions is essential to enhance the recruitment of student volunteers. Thirdly, outreach efforts to the community and the social and healthcare system are vital to raise awareness about the programme among the target population, healthcare professionals, and caregivers. Lastly, extensive lobbying and continuous dialogue with policymakers are required to prioritise equity-creating initiatives on the political agenda. 

The Social-Health Bridge-Building Programme may face challenges when considering broader implementation. Given its extracurricular status, the programme’s success hinges on effectively recruiting, training, and involving a significant number of student volunteers. In rural areas, this recruitment is particularly challenging due to the concentration of healthcare students in urban centres near their educational institutions. A potential remedy might be to incorporate the Social Health Bridge-Building Programme into the formal curriculum. However, this approach requires in-depth research, and a thorough understanding of educational paradigms. While such integration might alleviate some recruitment issues, it does not necessarily resolve the challenge of attracting student volunteers in rural settings.

### 4.3. Implications for Research 

Despite the growing societal and political attention given to social inequality within the healthcare system, there remains a significant research gap regarding the underlying mechanisms that contribute to social inequality in patient-healthcare system interactions. Our socio-ecological model of barriers to healthcare access provides a detailed examination of these mechanisms. However, several knowledge gaps still exist, limiting the insights that can be derived from existing scientific literature. More studies, particularly in healthcare settings with universal coverage are needed, focusing on investigating the causes of social inequities in healthcare encounters. Furthermore, there is a substantial lack of knowledge regarding interventions aimed at reducing social inequity in healthcare. Therefore, future research should prioritise exploring the impact of such interventions. As the implementation of the Social Health Bridge-Building Programme progresses, it is crucial to gain further insights into its effectiveness, implementation process, mechanisms of impact, and potential for scaling up. Consequently, a comprehensive evaluation of the programme, utilising both qualitative and quantitative research methods, is underway. This paper has the potential to inform future enhancements, modifications, and expansions of the programme, thus further addressing healthcare inequities.

### 4.4. Strengths and Limitations

This study offers valuable insights into a unique programme aimed at promoting equity in healthcare. The Social Health Bridge-Building Programme stands out from other existing programmes in several aspects. Firstly, while existing evidence primarily focuses on programmes developed and implemented in an American healthcare setting, this paper presents a programme developed and implemented in a setting with universal healthcare coverage. Secondly, the target population is not limited to disadvantaged individuals but includes a broader range of individuals. In a universal healthcare setting, access to healthcare is not directly influenced by income or occupation, as it might be in other countries with more private financing of healthcare, such as the United States. However, a significant group of individuals still encounter challenges in navigating the healthcare system. Therefore, the Social Health Bridge-Building Programme focuses on providing assistance in navigating the healthcare system and obtaining social support, rather than solely addressing the issue of legal access to healthcare or social resources. Additionally, unlike existing publications, student volunteers in this programme are not limited to medical students but include individuals from various educational backgrounds, ranging from nurses to public health students. Thirdly, the active engagement of student volunteers with individuals in vulnerable situations, accompanying them and being present at their bedside, enhances their ability to understand the patient’s perspective and exposes them to different ways of life. 

Some methodological strengths and limitations should be discussed. A major strength of this paper is the thorough use of evidence and the theory-driven elaboration within the context of the Social Health Bridge-Building Programme. The well-designed logic model, which addresses the programme’s impact at multiple levels (individual, student volunteers, and the healthcare system), is a noteworthy strength. This, together with the systematic and transparent documentation of programme details using the TIDieR checklist, enhances reproducibility and make the implementation of the Social Health Bridge-Building Programme in various contexts more feasible. The close collaboration with stakeholders involved in the programme enhances the overall credibility of this study. A limitation of this paper is that the programme theory was elaborated after the implementation of the programme and not during the programme’s initial development. If the programme theory had been developed at an earlier stage, it would have allowed the evidence review and theory to guide the programme’s development. Instead, empirical evidence and theory were employed to explicate and substantiate the assumptions regarding the causal pathways through which the desired effects of the programme are achieved. Nonetheless, this paper still provides crucial knowledge for establishing the foundation for future evaluations and implementation in other settings.

## 5. Conclusions

In conclusion, this study provides important insights into a unique programme aimed at promoting equity in healthcare while educating future healthcare professionals. Considering the contextual differences across healthcare settings, understanding the specific context in which a programme is implemented is crucial. As the implementation of the Social Health Bridge-Building Programme progresses, there is a need to gain further insights into its effectiveness, implementation and mechanism of impact, and potential for scaling up. Hence, a comprehensive evaluation of the programme covering qualitative and quantitative research methods is forthcoming. This paper can inform future improvements, modifications, and expansion of the programme to further address healthcare inequities.

## Figures and Tables

**Figure 1 ijerph-20-06837-f001:**
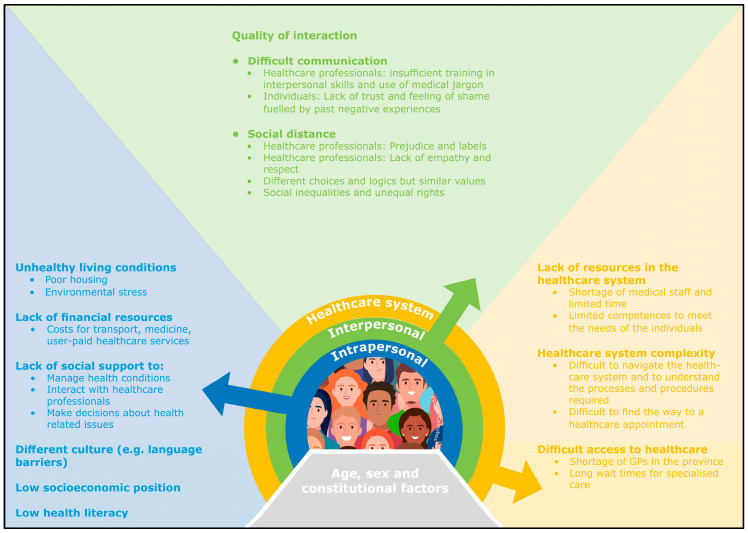
Socio-ecological model of barriers to access to and benefit from healthcare for individuals in vulnerable situations.

**Figure 2 ijerph-20-06837-f002:**
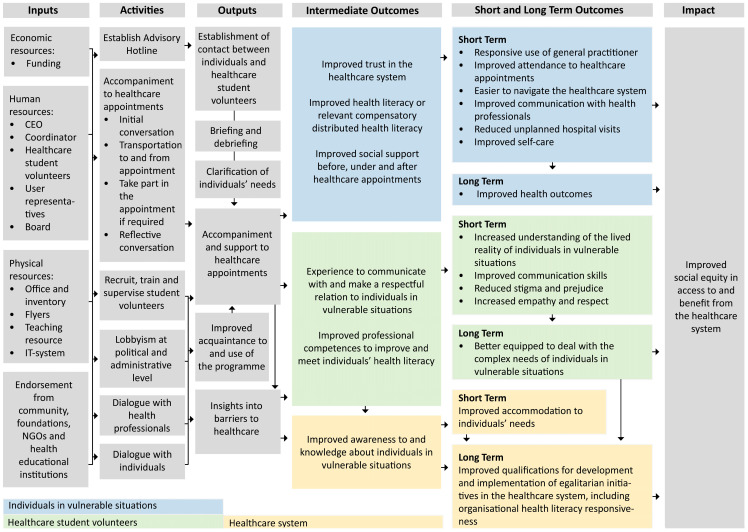
Logic model of the core resources, activities and output as well as the intended outcomes of the Social Health Bridge-Building Programme.

**Table 1 ijerph-20-06837-t001:** Systematic review summary table.

Author (Year)	Country of Origin	Target Population (n)	Student Volunteer (n)	Intervention	Design and Methods	Outcomes	Main Conclusions
Accompaniment and support in the Social Health Bridge-Building Programme							
Tonnesen (2023) [25]	Denmark	Persons in vulnerable situations (10)	Health care students (8)	The Social Health Bridge-Building Programme: Health care students accompany and support persons in vulnerable situations to health care appointments.	Case study Ethnographic fieldwork using interviews and participant observation in the form of “walking fieldwork”.	Important components in bridge-building Learning outcomes	Persons in vulnerable situations get social support to make it to health appointments. Safe-making and wayfinding are important components in bridge-building with bridge-builders acting as as-if-relatives. Bridge-building is an investment in future health professionals’ understanding of vulnerability in lives and barriers to health access.
Clinical work in student-run or shelter-based health clinics							
Asgary (2016) [46]	US	Homeless (? *)	Medical students (30)	Clinical exposure in shelter-based clinics	Case study Mixed methods: surveys, debriefing sessions and observed clinical skills	Learning outcomes	Medical students were better prepared to address the multi-level barriers to healthcare among homeless due to a clinical and population-based curriculum.
Wilson (2023) [45]	North America (n = 73), Australia (n = 6), Europe (n = 8), South Africa (n = 2), Brazil (n = 1), New Zealand (n = 1)	Underserved patients (? *)	Medical students (80%) Other students (20%)	Student-run clinics (SRCs)	Systematic review	Learning outcomes	SRC participation was linked with improved clinical skills (e.g., improved interpersonal skills including interpersonal communication skills and patient interaction relationship skills), interprofessional skills, empathy and compassion, in particular improving attitudes towards, awareness and understanding of the needs and social reality of the underserved patients, and leadership experience.
Screening of social needs in Community Resource Programmes							
Gautam (2022) [47]	US	Vulnerable people (791)	Medicine, nursing, undergraduate and graduate in general (24)	The Community Resource Navigator model: student volunteers help patients connect with community-based organisations.	Case study (Description and evaluation of curriculum for the Community Navigator Model)	Learning outcomes	Students’ gained knowledge on social determinants of health and health disparities of the local community and skills to address social needs. Students developed broader interpersonal and communication skills needed to become future health professionals.
Losonczy (2017) [48]	US	Patients at a safety net county-owned hospital (Intervention. 154 control: 305)	Undergraduate (Health science) (? *)	The Highland Health Advocates (HHA): Students placed in the emergency department screen patients for health-related social needs and help them navigate the system of social services.	Description and evaluation of the HHA model	Patients perspectives	More HHA subjects were linked to a resource (59% vs. 37%) and a medical home (92% vs. 76%.) At 6 months, 75% felt HHA was helpful and more HHA subjects had a doctor (93% vs. 69%). No difference was found in ED utilisation, primary need resolution or self-reported health status.
Onyekere (2016) [49]	US	High risk patients who have resource needs (369)	Osteopathic medicine (31)	The Medical Student Advocate (MSA) programme: students placed in coordination teams at a primary care practice serving a diverse patient population.	Case study Video reflection sessions	Learning outcomes	Increased empathy and understanding of the social determinant of health.
Sandhu (2021) [50]	US	Vulnerable people (26)	Medicine psychology and public policy (4)	The Community Resource Navigator model: student volunteers help patients connect with community-based organisations.	Case study/Pilot study (description of the programme development and evaluation of feasibility)	Feasibility (e.g., # of patients connected to prescribed resource)	Student volunteers are untapped resources to support integrated health and social care.

* No information on the number of individuals in vulnerable situations or the number of student volunteers included in the study.

## Data Availability

Data sharing not applicable.

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
