# Peer review of "Bridging Inequity Gaps in Healthcare Systems While Educating Future Healthcare Professionals—The Social Health Bridge-Building Programme"

_ijerph, 2023, doi:10.3390/ijerph20196837_

Round 1

Reviewer 1 Report

Thank you for the opportunity to review this engaging and well written manuscript. I have only a few comments and suggestions.

Introduction: Can you address any of the upstream factors (e.g. racism, discrimination) that may lead to social inequalities?

2. Materials and Methods:

3.1: Identifying Barriers: were there any other barriers that were identified? Again, I am wondering about upstream mechanisms like medical mistrust, lack of access to education and employment, housing instability, different culture—but compared to what? The majority or dominant culture? Why may someone have low health literacy? I would encourage you to think about the systemic impact of why these things happen so that health care providers and health systems can see that these inequities exist in a complex relationship with systemic oppression and are not just based on individual choices. You describe them in the text but the graphic is slightly reductive.

Overall, this is an excellent paper that describes an interesting and potentially highly adaptable program.

Reviewer 2 Report

The topic proposed by the authors is of potential interest. Doubts remain about the systematic review performed. Primarily, what are generic free text search terms? They are not clearly stated in the text. The screening of the literature was performed by one or two investigators? In blind? The authors are asked to check the selected works and better describe the excluded ones. Also, some recent articles may be added (doi:10.3389/fpubh.2022.870386; doi:10.3389/fpubh.2022.869793; doi:10.1002/cl2.1237). The opportunities offered by digitalization and new technologies, such as artificial intelligence (AI) and robotics, are a lever for better communication, the removal of barriers and for making social welfare services more efficient for vulnerable people. Again, in this field there are several articles that could be considered and included. The authors have included a subparagraph “Strengths and limitations”. However, the application limits of the Social Health Bridge-Building Program are not described at all; this part is crucial and the authors should provide more details.

Reviewer 3 Report

Review for Int. J. Environ. Res. Public Healt

Bridging inequity gaps in healthcare systems while educating future healthcare professionals – The Social Health Bridge- Building Programme

This article seeks to describe an intervention to reduce health care inequity in a universal insurance (ie non US) context.  It nicely presents the Social Health Bridge-Building Programme employing the Template for Intervention Description and Replication (TIDieR) reporting process.

Overall the paper, although not a completed study, lays out the model and theory of a program that has been implemented across several Danish cities and will be evaluated.  It is novel in that the intervention includes students beyond medical students.  It intends to shape the sympathies and skills of students before they formally enter the health care system for their careers –be it nursing, physician, psychologist, social work.

Comments-

What is going on with the page numbering?

Mild editing for subject verb (ie Singular/plural) agreement (Singular/plural)

The strengths and limitations of this paper need further development.  For example, a strength is that the use the TIDieR method to describe the program, the ogic model is well-design and address all three level of the intervemntion (vulnerable population, volunteer and health system)

Programme yet to be studied—model includes volunteers beyond med students (not yet a strength)

As noted, mild editing for subject verb (ie Singular/plural) agreement (Singular/plural) would improve the paper

Round 2

Reviewer 2 Report

The manuscript has been significantly improved and the required points have been well addressed by the authors. Improving equity by addressing individual, social and systemic barriers to healthcare also targets people with mental disabilities. You clearly stated that the paths proposed by the program's activities towards greater equity in healthcare need to be clarified and updated. Artificial intelligence and digital systems can provide important help in this regard. However, regarding mental disorders, in addition to the articles already previously suggested, please implement the number of references (e.g., doi:10.1371/journal.pone.0258729; doi:10.1186/s12909-019-1472-7).
